# Improved Expressivity Through Dendritic Neural Networks

**Xundong Wu**     **Xiangwen Liu**     **Wei Li**     **Qing Wu**
School of Computer Science and Technology
Hangzhou Dianzi University, Hangzhou, China
wuxundong@gmail.com, wuq@hdu.edu.cn

A typical biological neuron, such as a pyramidal neuron of the neocortex, receives thousands of afferent synaptic inputs on its dendrite tree and sends the efferent axonal output downstream. In typical artificial neural networks, dendrite trees are modeled as linear structures that funnel weighted synaptic inputs to the cell bodies. However, numerous experimental and theoretical studies have shown that dendritic arbors are far more than simple linear accumulators. That is, synaptic inputs can actively modulate their neighboring synaptic activities; therefore, the dendritic structures are highly nonlinear. In this study, we model such local nonlinearity of dendritic trees with our dendritic neural network (DENN) structure and apply this structure to typical machine learning tasks. Equipped with localized nonlinearities, DENNs can attain greater model expressivity than regular neural networks while maintaining efficient network inference. Such strength is evidenced by the increased fitting power when we train DENNs with supervised machine learning tasks. We also empirically show that the locality structure of DENNs can improve the generalization performance, as exemplified by DENNs outranking naive deep neural network architectures when tested on classification tasks from the UCI machine learning repository.

## 1   Introduction

Deep learning algorithms have made remarkable achievements in a vast array of fields over the past few years. Notably, deep convolutional neural networks have revolutionized computer vision research and real-world applications. Inspired by biological neuronal networks in our brains, originally in the form of a perceptron [37], an artificial neural network unit is typically constructed as a simple weighted sum of synaptic inputs followed by feeding the summation result through an activation function. Typical neural network units can be depicted as $\sigma(\sum_{i=1}^{m} w_i x_i)$ and exemplified as components in Fig. 1(a). Such a scheme has been used in almost all modern neural network models.

In the physiological realm, however, as revealed by both experimental and modeling studies, biological neurons are way more complicated than the simple weighted sum model described above. It is shown that dendritic arbors (see Fig. A1 in appendix A for a schematic diagram of neurons and their dendrite arbors) contain an abundance of ion channels that are active, in other words, superlinear [41, 24, 23, 39, 25]. The existence of those active channels, combined with the leaky nature of dendrite membrane, suggests that one synaptic input can have nonlinear influence on synapses that are in its close proximity. In addition, studies on synapse plasticity also show strong evidence that the biological machinery for plasticity also acts locally inside dendrites [24, 2]. Such properties greatly elevate the contribution of local nonlinear components in neuronal outputs and empower neuronal networks with much greater information processing ability [41, 17, 33, 25, 45, 32].

Despite the continual progress of deep learning research, the performance of artificial neural network models is still very much inferior to that of biological neural networks, especially in small data learning regimes. Naturally, we want to return to our biological brains for new inspirations. Could the active dendrite structure be part of the inductive bias that gives our brains superior learning ability? Here, we introduce a neural network structure that aims to model the localized nonlinearity and

plasticity of dendrite trees and explore the advantages that active dendrites can bring to supervised learning tasks.

To extract the local and nonlinear nature of dendritic structures, we design our dendritic neural network (DENN) model as shown in Fig. 1(b). Constructed to replace a standard, fully connected, feedforward neural network (FNN) as in Fig. 1(a), every neuron in this model also receives a full set of outputs from the earlier network layer or input data as in a standard FNN. However, in this case, the connection maps at the dendrite branch level are sparse. The synaptic inputs are first summed at each branch, followed by nonlinear integration to form the output of a neuron (see section 3 for detail). We also ensure that learning events are isolated inside each branch for every pattern learning.

We test DENN models on typical supervised machine learning tasks. It is revealed that our DENN structure can give neural network models a major boost in expressivity under the fixed parameter size and network depth. At the same time, it is empirically shown that the DENN structure can improve generalization performance on certain machine learning tasks. When tested on 121 UCI machine learning repository datasets, DENN models outrank naive standard FNN models.

Our proposed DENN structure also avoids the typical computing inefficiency associated with sparse neural networks and can enable efficient sparse neural network inference. This structure, which we call intragroup sparsity, can potentially be adopted in general deep neural networks.

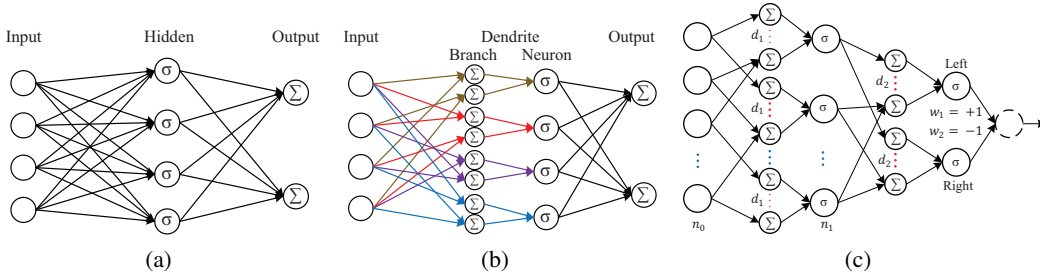

Figure 1: (a) A simple standard FNN with one hidden layer; (b) A DENN structure; (c) This DENN is a universal approximator. The hidden layer in the standard FNN is decomposed into two stages for the DENN. At the first stage, the dendrite branch performs the linear weighted sum of the sparsely connected inputs. At the second stage, the outputs of all branches are nonlinearly integrated to form the neuron output. The last layer of the DENN is kept same as that in a regular FNN.

## 2   Related Work

**Dendritic neural networks:** Driven by the disparity between classic artificial neural networks (ANNs) and biological neuronal networks, a notable amount of works have been done on integrating nonlinear dendrites into ANNs. Computational neuroscience research indicates that the nonlinear dendritic structure can strengthen the information processing capacity of neural networks [41, 32, 33, 45, 17, 13, 9, 38]. In [32], the authors map pyramidal neurons onto two layer neural networks. In the first layer, the sigmoidal subunits are driven by synaptic inputs. In the second layer, the subunit outputs are then summed and thresholded by the cell output. Poirazi and Mel [33] show that the dendritic structures can access a large capacity by a structural learning rule and random synapse formation. In [35] the authors introduce and develop a mathematical model of dendritic computation in a morphological neuron based on lattice algebra. Such neurons with dendrites are able to approximate any compact region in Euclidean space to within any desired degree of accuracy. Hussain *et al.* [13] propose a margin based multiclass classifier using dendritic neurons, in which the value of synapses is binary. Their networks learn the data by modifying the structure, the connections between input and dendrites, not the weights. The network used in our study is different from earlier studies in that our network is structured to abstract both the nonlinearity and localized learning nature of dendritic arbors. And our network architecture is designed with inference efficiency in mind. We also apply this model to typical machine learning tasks with good results.

**Deep neural network:** The remarkable success of the deep feedforward neural networks has driven many researchers attempting to understand the reason behind such achievement. One essential factor, the high expressive power of a deep network, is believed to underlie their success. In [34, 27, 26, 36], the authors study the neural network expressivity by measuring the number of linear regions or transition events between linear regions in the function space. A DENN can also be considered as a network with additional depth. In this study, we also investigate the expressive power of DENNs, specifically, how the size of dendrite branches affects the DENN performance.

**Sparse neural network:** There have been many works on the sparse neural networks[4, 22, 7, 20, 29]. Sparsity can endow neural network models with the advantage of lower computational, communication, and memory requirements. In a large learning system, such as the human brain , which contains $10^{11}$ neurons, sparse networks are the only design that is feasible. The very successful convolutional neural network can also be considered as a sparse network with synaptic connections restricted to local neighborhoods. The sparsity of neural networks can be built by direct construction, by sparsity inducing regularization [40], or by post learning weight pruning [11, 21, 44]. Our model belongs to the category of sparsity through construction. Several important features make our model to stand out from other sparse models. First, of all our model is actually not sparse at the neuron level; sparsity only appears at the dendrite level. Second, practically no extra space is required to store the connection maps of DENNs. Furthermore, the regularity structure associated with DENNs can enable efficient computing of our networks as contrast with typical sparse neural networks. We will elaborate on those properties in the following sections.

**Learned piecewise activation functions:** Learned piecewise activation functions [28, 8, 10] have been the essential building blocks for the recent success of deep neural networks. Most relevant to this work, in [8], the authors propose to apply the Maxout output function for piecewise linear neural networks with enhanced performance. In our model, we adopt the Maxout function to enable localized nonlinear synaptic integration and divisional learning units. However, our model is very different from the original Maxout network since every branch in a dendrite of our model receives mutually exclusive inputs.

**Random feature subsets:** In our model, the inputs to a specific dendrite branch are randomly selected. Such an approach resemble the practice in traditional machine learning algorithms that use feature subsets to build an ensemble of models [49, 3]. The model ensemble built in such an approach can give models better generalization performance. In our model, feature subsets are used to construct features of dendrite subunits of hidden layers.

## 3 Dendritic Neural Network

### 3.1 Definitions

A standard *feedforward neural* network (FNN) defines a function $F : \mathbb{R}^{n_0} \to \mathbb{R}^{out}$ of the form

$$F(x) = f_{out} \circ f_L \circ \cdots \circ f_1(x).$$

A layer $l$ of a standard feedforward neural network consists of computational units that define a function $f_l : \mathbb{R}^{n_{l-1}} \to \mathbb{R}^{n_l}$ of the form

$$f_l(x_{l-1}) = [f_{l,i}(x_{l-1}), \dots, f_{l,n_l}(x_{l-1})]^\top,$$
$$f_{l,i}(x_{l-1}) = \sigma_r(W_{l,i}x_{l-1} + b_{l,i}), i \in [n_l],$$

where $f_{l,i} : \mathbb{R}^{n_{l-1}} \to \mathbb{R}$ is the function of the $i^{th}$ output unit in layer $l$, $W_l \in \mathbb{R}^{n_l \times n_{l-1}}$ is the *input weight matrix*, $x_{l-1}$ is the *output vector* of layer $l-1$ and the *input vector* of layer $l$, $b_l \in \mathbb{R}^{n_l}$ is the *bias vector*, $\sigma_r$ is the *activation function* for each layer $l \in [L]$, $L$ is the number of layers of a network and $\{1, 2, \dots, L\}$ is denoted by $[L]$. We mainly consider rectifier units $\sigma_r(x) = \max\{0, x\}$.

In comparison, a DENN is a composition of *dendritic layers* that defines a function $F_D : \mathbb{R}^{n_0} \to \mathbb{R}^{out}$ given by

$$F_D(x) = f_{out} \circ f_L^D \circ \cdots \circ f_1^D(x),$$

where $f_l^D : \mathbb{R}^{n_{l-1}} \to \mathbb{R}^{n_l}$ is the function of the dendritic layer $l$. Layer $l$ has $n_l$ *dendrite units* and each dendrite unit is associated with one *neuron output*, which is the maximum of $d$ *dendritic branches*. Each branch makes connections with $k$ inputs from the last layer. The selection strategy is

that each branch randomly chooses $k = n_{l-1}/d$ connections without replacement from $n_{l-1}$ inputs. This strategy assures that every input feature is only connected to a dendrite unit once to avoid redundant copies and that synapse sets of each branch are mutually exclusive for each neuron. Such a selection strategy is also backed up by some physiology studies that suggest axons avoid making more than one synapse connection to the dendritic arbor of each pyramidal neuron [47, 5]. The mutually exclusive connection selection strategy is also the basis for the efficient network inference.

Note that we should let $k < n_{l-1}$, that $k$ is called the *branch size*, and that $d$ is called the *branch number*. The branch $h_{l,i,j}$ is given by $h_{l,i,j} : \mathbb{R}^{n_{l-1}} \to \mathbb{R}$ of the form

$$h_{l,i,j}(x_{l-1}) = \sum_{m=1}^{n_{l-1}} ((S_{l,i,j,m} \cdot W_{l,i,j,m})x_{l-1,m}), j \in [d],$$

where $W_l \in \mathbb{R}^{n_l \times n_{l-1} \times d}$ are the *weight matrices* between input units and branches, $S_l \in \mathbb{R}^{n_l \times n_{l-1} \times d}$ are the *mask matrices* to represent whether a branch has a connection with an input and the value of $S_l$ is binary (1 or 0). $S_l$ are generated with a deterministic pseudo random number generator from a preassigned random seed. That is, we can reproduce $S_l$ just from the original random seed. Therefore, with a proper algorithm, $S_l$ **will not incur extra model storage and transfer cost.**

The output of each neuron $g_{l,i}$ can be formulated by $g_{l,i} : \mathbb{R}^d \to \mathbb{R}, g_{l,i}(x_{l-1}) = \max(h_{l,i,j}(x_{l-1})) + b_{l,i}, i \in [n_l], j \in [d]$, where $b_l \in \mathbb{R}^{n_l}$ is the *bias vector*. A dendritic layer output $f_l^D$ is the composition of branches and can be formulated by

$$f_l^D(x_{l-1}) = [g_{l,1}(x_{l-1}), \ldots, g_{l,n_l}(x_{l-1})]^\top.$$

In particular, when $d = 1$, the dendritic layer is the same as an affine fully connected layer with linear activation functions. The output layer $f_{out}$ of a dendritic neural network is the same as the output layer of a feedforward neural network. For some cases, we use a fully connected layer with nonlinear activation functions as the first layer of DENN.

## 3.2 A Universal Approximator

A feed-forward network with one hidden layer of finite number of units is a universal approximator [12]. As shown in [8], a Maxout neural network with two hidden units of arbitrarily many components can also approximate any continuous function. Similarly, a DENN can also be a universal approximator. We have a DENN with an input layer of $n_0$ units, the first layer has $n_1$ output units and $d_1 \le n_0, d_1 \in \mathbb{N}^+$, the second layer has two output units and $d_2 \le n_1, d_2 \in \mathbb{N}^+$. Then, we use a virtual unit to produce the difference of the two output units as shown in Fig. 1(c). Now consider the continuous piecewise linear (PWL) function $g(x)$ and the convex PWL function $h(x)$ consisting of $d_2$ locally affine regions on $x \in \mathbb{R}^n$. Each affine region of $h(x)$ is determined by the parameter vectors $[W_i, b_i], i \in [1, d_2]$. Then, we prove that a two-layer DENNs can approximate any continuous function $f(x), x \in \mathbb{R}^n$ arbitrary well with sufficiently large $n_1$ and $d_2$.

**Proposition 1** *(Wang[43]): For any positive integer $n$, there exist two groups of $n + 1$ dimensional real-valued parameter vectors $[W_{1i}, b_{1i}]$ and $[W_{2i}, b_{2i}], i \in [1, d_2]$ such that $g(x) = h_1(x) - h_2(x)$. That is, any continuous PWL function can be expressed as the difference of two convex PWL functions. [8, 43].*

**Proposition 2** *Stone-Weierstrass Approximation Theorem: Let $C$ be a compact domain $C \in \mathbb{R}^n$, $f : C \to \mathbb{R}$ be a continuous function, and $\epsilon > 0$ be any positive real number. Then, there exists a continuous $PWL$ function $g$, such that for all $x \in C, |f(x) - g(x)| < \epsilon$. [8].*

**Theorem 3** *Universal Approximator Theorem: Any continuous function $f$ can be approximated arbitrarily well on a compact domain $C \subset \mathbb{R}^{n_0}$ by a two-layer dendritic neural network, with $d_1 = 1$ and $n_1$ output units in the first layer and $d_2 \le n_1, d_2 \in \mathbb{N}^+$ in the second layer with sufficiently large $n_1$ and $d_2$.*

**Proof Sketch:** Let there be a two-layer dendritic neural network with the first layer of $n_1$ output units and $d_1 = 1$. The second layer has two output units, and $d_2 \le n_1, d_2 \in \mathbb{N}^+$. Then, we construct a virtual unit to output the final result, the difference of the two output units. An output unit in the first layer is a hyperplane in $\mathbb{R}^n$ and a branch in the second layer is also a hyperplane because it is the

sum of $n_1/d_2$ hyperplanes. An output unit in the second layer is the maximum of $d_2$ branch units. In other words, the output unit is the upper envelope of $d_2$ hyperplanes, and it represents a convex PWL function. From Proposition 1, any continuous PWL function can be expressed by the virtual unit, a difference of two convex PWL functions. Then, according to Proposition 2, any continuous function can be approximated arbitrarily well by this network to achieve the desired degree of $\epsilon$ with sufficiently large $n_1$ and $d_2$ on the compact domain $C$. Consequently, a two-layer neural network satisfies Theorem 3 and then the Theorem holds. In general, as $\epsilon \to 0$, we have $n_1, d_2 \to \infty$.

Theorem 3 is restrictive with $d_1$ limited to 1. Here, we extend it to allow $d_1 \leq n_0$ with $d_1 \in \mathbb{N}^+$.

**Theorem 4** *Generalized Universal Approximator Theorem: Any continuous function $f$ can be approximated arbitrarily well on a compact domain $C \subset \mathbb{R}^{n_0}$ by a two-layer dendritic neural network, with $d_1 \leq n_0$ and $n_1$ output units in the first layer and $d_2 \leq n_1, d_2 \in \mathbb{N}^+$ in the second layer with sufficiently large $n_1$ and $d_2$.*

**Proof Sketch:** From [8] we know that a Maxout neural network with two hidden units of arbitrarily many components can approximate any continuous function. Assume that such a Maxout neural network approximates the target function with $m$ components in each hidden unit. We note two hidden Maxout units as the left unit and the right unit. We prove Theorem 4 by constructing a network that is equivalent to such a Maxout neural network. We set $d_2 = m$, and the second layer dendrite size $k = n_0$. From there we arrive at $n_1 = 2n_0 m$. We assume all branches of the two second layer neurons receive mutually exclusive inputs from first layer neurons. This can be simply achieved through separating the first layer neurons into two independent groups as the second layer. In this way, every branch of the second layer neuron receives inputs from exactly $n_0$ first layer neurons. We denote the first layer input neurons as $h_{ij}$, with $i \in \{1, 2, \ldots, 2m\}$ indexes the branches of second layer neurons, $j \in \{1, 2, \ldots, n_0\}$. Denote the input of the network as $x_0 k$. For every first layer neuron $h_{ij}$, we set all input weights in those neurons to 0, except the weight connection from $x_0 k$ when $k = j$. The biases of branches of all zero weights are then set to negative infinity. The bias of the branch of the nonzero weight are then set to zero. With this construction, each branch of the DENN second layer neurons receive a full set of $n_0$ network inputs and weights corresponding to each inputs that are independent of other branches. Therefore, we have the equivalent network to the original Maxout network. Thus, we prove that a general two-layer DENN is a universal approximator.

### 3.3 Intragroup sparsity

Inside a DENN layer, the connection map between the layer inputs and the dendritic arbors are generated randomly and are sparse. Because of the poor data locality nature of generic sparse network structures, performing inference with those networks is generally inefficient [44]. We propose a design called intragroup sparsity. In this design, every DENN weight is given a single branch index, which tells us to which branch the weight feeds to. With this arrangement, each weight is first multiplied with the corresponding layer input as usual followed by the accumulation procedure directed by the branch index. That is, the multiplication result is added to a branch sum addressed by the index. Given the number of branches used in our model is typically small, we will only need a few bits to encode such an index. In this way, the intragroup sparsity structure avoid the poor data locality issue associated with typical sparse neural networks.

### 3.4 Network Complexity

A wealth of literature has argued that deep neural networks can be exponentially more efficient at representing certain functions than shallow networks [27, 34, 30, 31, 26, 36]. In [27], the author proposes that deep neural networks gain model complexity by mapping different parts of space into same output. A network gains exponentially efficiency through the reuse of such maps of different filters of later layers. That is, the neural networks having more linear regions can represent functions of higher complexity and have strong expressive power. More specifically, a perceptron with the ReLU nonlinearity can divide input space into two linear regions by one hyperplane. A DENN neuron with $d$ branches contains $h = d(d-1)/2$ divisional hyperplanes. According to the Zaslavsky theorem [48] those hyperplanes can then divide the input space into a number of regions bounded above by $\sum_{s=0}^{n_0} \binom{h}{s}$. Empirical tests indeed show that DENNs can have boosted model complexity,

as highlighted by the number of region transitions (see Appendix B). Can such model complexity translate into better model fitting power? We test this hypothesis in the next section.

## 4    Result

In the last section, we show that a DENN can divide function space into more linear regions than a standard FNN. We would like to verify whether dividing function space into a greater number of regions actually translates into greater expressive power when the DENN is used to model real world data. We also would like to determine if the inductive bias, an imitation of biological dendritic arbors, built into the DENNs can be beneficial for machine learning tasks, i.e., improved generalization performance. Hence, we empirically compare the performance of DENNs with several FNN architectures on permutation-invariant CIFAR-10/100 and Fashion-MNIST [46] data and 121 machine learning classification datasets from the UCI repository to find out.

### 4.1    Permutation-invariant image datasets

We first test our models on permutation-invariant image datasets: the Fashion-MNIST[46], CIFAR-10 and CIFAR-100 datasets[18]. The Fashion-MNIST dataset consists of 60,000 training and 10,000 test examples and each example is a $28 \times 28$ pixel grayscale article image, labeled with one out of ten classes. In this experiment, the inputs of the network are of $28 \times 28 = 784$ dimensions. The CIFAR-10/100 dataset consists of 50,000 training and 10,000 test images, and each image is a $32 \times 32$ color image, associated with one of 10 or 100 classes respectively. For permutation-invariant learning, models should be unaware of the 2-D structure of images used in learning.

The baseline standard FNN used in this experiment is composed of three layers with $n$ units for the first two layers and 10/100 category outputs for last layer. The output function used for the hidden units is the popular ReLU function [19]. For the network outputs, the Softmax function is used to generate the network outputs. The cross entropy loss is then calculated between the Softmax output and ground-truth labels. To facilitate model training, we also insert batch normalization [14] or layer normalization [1] in the first two layers of the standard FNNs (Indeed, we are giving control models an extra advantage here.). We construct a DENN and a Maxout network to compare against the standard FNNs. For a proper comparison, we keep the number of synaptic parameters constant across different architectures. That is, for DENNs, the number of hidden units $n$ are kept the same as in the baseline models. When we vary the number of dendrite branches $d$ in each neuron, the number of synaptic weights is set to $n/d$ to keep the number of synapses the same as in the standard FNN hidden layer. For the Maxout network, by increasing the number of kernels in a Maxout unit, the number of units is reduced accordingly to keep the total number of synaptic parameters constant. The output layers of the DENN and Maxout models are kept same as those of the standard FNNs.

We train all models in comparison with the Adam optimizer [15] for 100 epochs. The learning rate used is decayed exponentially from 0.01 to $1e-5$ unless otherwise stated. To show the model fitting power at plain state, we do not apply any kind of model regularization in our model training. For this test, data are preprocessed with per-image standardization.

We start by training models on Fashion-MNIST and CIFAR-10 datasets to test the fitting power of models with layer size $n$ set to $512$. As shown in Fig. 2, DENNs can attain much lower training loss than other network architectures on both datasets. More specifically, in Fig. 2(a) and Fig. 2(c), the lowest training loss values of the DENNs and Maxout networks of different branch numbers $d$ (kernel number for Maxout) are compared against the lowest loss from standard FNNs with batch normalization or layer normalization (the loss of regular FNN without normalization and self-normalizing neural networks (SNNs) are much higher and not shown). As we can observe, DENNs with mid-size dendrite branch numbers can attain much lower loss than standard FNNs. The performances of the DENNs with extreme branch numbers, for example, 2 and 256 branches are comparable with standard FNNs. Regular Maxout models are known to benefit from their extra parameters from additional kernels. In this experiment, constrained by the constant synaptic weight number requirement, the performance of Maxout networks deteriorates as we decrease the number of hidden units to compensate for the increased number of kernels per unit. Typical sets of training loss curves are also shown in Fig. 2(b) and Fig. 2(d) respectively. Clearly, DENNs attain much lower training loss than other FNNs. We did not show the training accuracy curves here because they tend to reach quite close to 100% accuracy due to overparameterization and thus are not truly informative.

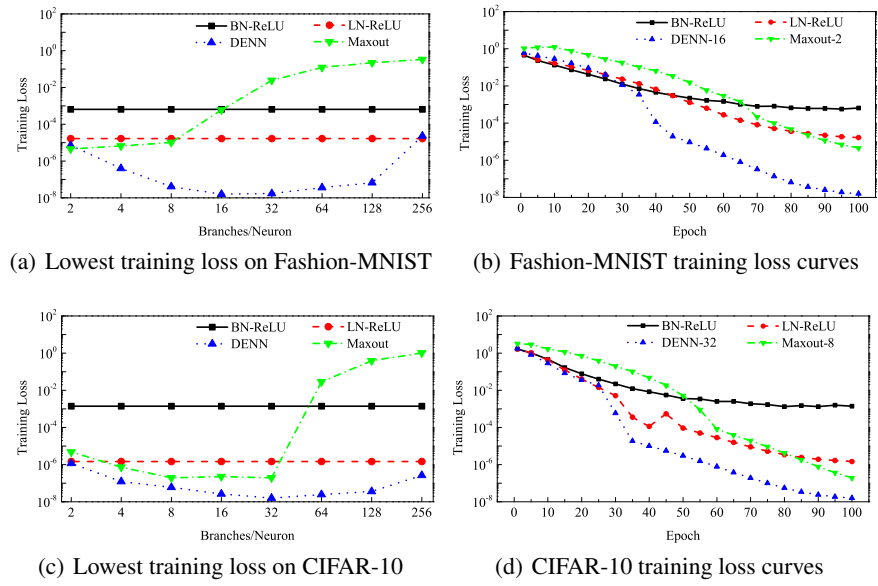

(a) Lowest training loss on Fashion-MNIST      (b) Fashion-MNIST training loss curves

(c) Lowest training loss on CIFAR-10      (d) CIFAR-10 training loss curves

Figure 2: Training loss results on Fashion-MNIST and CIFAR-10 dataset for ReLU FNNs (batch normalization-ReLU (BN-ReLU) and layer normalization-ReLU (LN-ReLU)), DENNs and Maxout networks. The Y axis is set to the logarithmic scale.

To obtain a better understanding, we perform further experiments to test models with different model sizes where they are under more capacity pressure.

In Fig. 3 we show the results from training models on CIFAR-100 dataset with layer size $n$ set to 64, 128, 256 and 512 respectively. DENNs show clear advantage over other FNN models when under capacity pressure observable on both training accuracy and loss curves.

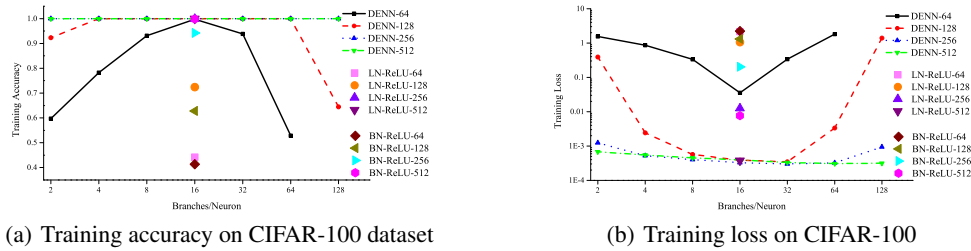

(a) Training accuracy on CIFAR-100 dataset      (b) Training loss on CIFAR-100

Figure 3: Training accuracy and training loss results on CIFAR-100 dataset for DENNs with different branch numbers along with results of ReLU FNNs (batch normalization-ReLU i.e. BN-ReLU and layer normalization-ReLU i.e. LN-ReLU).

We also perform experiments on DENNs with two different nonlinear activation functions other than the Maxout function. In the first architecture, synaptic inputs are first summed, followed by passing individual dendrite summation result through the ReLU function. Then, we sum the dendritic outputs and pass it through a secondary ReLU function. The second architecture differs from the first architecture in that no secondary ReLU is used. The DENN with Maxout nonlinearities clearly outperforms the two ReLU-based architectures(see Appendix C, Fig. A10-13). It is worth to mentioning that even though those two architectures under-perform the standard DENNs, they also show rather interesting learning behaviors. We will further explore on this in our future work.

To obtain a better understanding of the learning behaviors of DENNS, we also train networks data synthesized from a random Gaussian noise generator (See Fig. A14 of Appendix C). Learning data with random data means models are less likely to take advantage of the redundancy in the data and thus require more model capacity. DENN models appears to hold less of an advantage over standard

FNN than when they are trained with regular image data. This result might indicate that part of the superior performance of DENN over standard FNNs comes from taking advantage of the correlation inside the data [22, 4].

Additional experimental results on permutation-invariant image datasets, including results on the test sets, can be found in Appendix C.

## 4.2 121 UCI Classification Datasets

In addition to evaluating the fitting power of DENNs on image datasets, we also evaluate the generalization performance of our DENNs on a collection of 121 machine learning classification tasks from the UCI repository as used in [16, 6, 42]. The data collection covers various application areas, such as biology, geology or physics. We compare the generalization performance of DENNs to those of standard FNN architectures tested in [16] on the dataset collection for its generalization performance. To obtain the benchmark result of a specific standard FNN method, a grid search is performed to search for the best architecture and hyperparameters with a separate validation set. The best model is then evaluated for accuracy on the predefined test set. We repeat the experiment on self-normalizing neural networks (SNNs) and obtain very similar results as in [16]. For all standard FNN architectures other than DENNs, we use the benchmark results that are included in [16] for the rest of the experiments. A detailed list of test results on those standard FNN networks and the list of their corresponding grid search hyperparameter space can be found in the Appendix A.4 of [16].

For this part of the experiment, the DENNs used are all composed of three network layers. For the first layer, a fully connected ReLU input layer is used to accommodate a wide range of input feature dimensionality. The second layer is a DENN layer, followed by the third layer, a fully connected softmax output layer. The width of the input layer and the DENN layer are both set to 512 units. The output layer width is set to equal to the number of classes for the corresponding dataset. For the DENN layer, we optimize the model architecture over the number of dendritic branches $d$ for each hidden unit. The value of $d$ is set to be one of $2^1, 2^2, \cdots, 2^8$ for each model. Correspondingly, the number of active weights in each dendrite branch is set to $512/d$. In addition to the regular DENNs models, we also train models with a dropout rate of 0.2. In total, for each task, we perform grid-search on 16 different settings.

As in [16], we train each model for 100 epochs with a fixed learning rate of 0.01. Different from [16], we use the Adam optimizer instead of the SGD optimizer for model training because DENNs are generally harder to train due to its sparse structure. For comparison, we also run tests of training SNN networks with the Adam optimizer, which gives worse results than the one trained with SGD optimizer. For the rest of the experiment, we follow the same procedure as in [16].

The accuracy results obtained on the DENNs can be found in Table A1. We ranked DENNs and other standard FNN architectures by their accuracy for each dataset and compare their average ranks(Table 1). DENNs outperform other network architectures in pairwise comparisons (paired Wilcoxon test across datasets). The comparison results are reported in the right column of Table 1.

Table 1: Comparison of DENNs with seven different FNN architectures on 121 UCI classification tasks. The 1st column lists the name of architectures, the 2nd column shows the average rank, and the 3rd column is the p-value of a paired Wilcoxon signed-rank test. Asterisks denote for statistical significance. DENNs get the best rank among network models in comparison.

| Method | Avg.rank | p-value | Method | Avg.rank | p-value |
|---|---|---|---|---|---|
| DENNs | 3.08 | | Highway | 4.38* | 9.08e-05 |
| SNN | 3.50 | 1.37e-01 | ResNet | 4.55* | 1.16e-05 |
| MSRAinit | 4.05* | 2.53e-03 | BatchNorm | 4.84* | 7.64e-07 |
| LayerNorm | 4.10* | 2.50e-03 | WeightNorm | 4.86* | 3.14e-07 |

## 5 Discussion

In this paper, we have proposed the DENN, a neural network architecture constructed to study the advantage that the dendrite tree structure of biological neurons might offer in learning. Specifically, we apply DENNs on supervised machine learning tasks. Through theoretical and empirical studies,

we identify that DENNs can provide greater expressivity than traditional neural networks. In addition, we also show that DENNs can improve generalization performances for certain data, as evidenced by the improved test accuracy on a large collection of machine learning tasks. Our DENN model is also designed to allow efficient network inference, which is not generally accessible to the typical sparse neural networks due to their built-in irregularity. In a DENN, each connection weight is associated with one and only one low bit-width branch index. Such regularity in its structure will allow efficient network computing with properly designed software and/or hardware. While such a sparse structure, noted as intragroup sparsity, was designed for DENNs, it can also be extended beyond DENNs to allow multiple network units to share one set (or a few sets) of inputs and thus enable a novel kind of inference efficient sparse network architecture.

We recall that DENNs can have diverse expressive power across different branch sizes when tested on permutation-invariant image datasets. Here, we give tentative discussion on the reason why DENNs have such behaviors. According to Theorem 3, any continuous function can be approximated by a continuous PWL function, which can be expressed as a difference of two convex PWL functions. A dendrite unit is the upper envelope of $d$ real-valued linear functions. The approximation error $\epsilon$ between the objective function and the output of the dendrite unit goes to zero, as $d \to \infty$. However, this result relies on an implicit condition that each linear function is without any constraint. As each dendrite branch connects with fewer inputs, the constraint for the linear functions would be stricter. The approximation error between the dendrite unit and the requested convex function would go up. Such a trade-off between the branch size and the branch number leads to optimal expressivity at mid-range dendrite sizes.

Thus far, we have not observed a clear advantage from DENN models in generalization performance on image datasets we tested on (see Appendix C for results on test set accuracy). It is possible we can obtain superior model generalization performance on those data by imposing additional regularization. Given the operating space granted by the substantially greater fitting power of DENN models, obtaining better generalization performance is certainly a possibility. We will test this in our follow-up works. Another possibility is to explore the nonlinear functions other than Maxout and ReLU we tested.

The network architecture used in this study is a boiled-down version of the biological neuronal network. DENNs model the essence of dendrite structure through enforcing localized nonlinearities and compartmentalized learning in dendrite branches. Such structure certainly does not cover the full complexity of dendrite arbors. In reality, we know that the influence of the activity of a certain synapse over its neighborhood decays quickly over the length of the branch. Such influence is also known to be modulated by meta-plasticity learning. In addition, the molecular machinery underlies synaptic plasticity relies on the diffusion of ion channels and signal molecules. Therefore, in the aspect of plasticity, the idea of treating a single dendritic branch as a learning compartment is another simplification. We also know that the size of dendritic branches can vary significantly, which can have compelling influence over the synaptic activation integration and localized plasticity inside dendrites. In this study, we used fixed sparsity masks that are generated at the initialization stage of neural networks. Such approach offers advantage that virtually no extra space is required to store the mask. As revealed by many earlier studies [33, 11], structural plasticity, i.e. changing the connection map can greatly improve model capacity. We will explore on this topic in our future work.

The complexity of biological neurons suggests we might need a learning apparatus with more built-in mechanisms or inductive bias to improve our neural network models. The following is a potential avenue that can be explored in our future research: To train a neural network with more than tens of thousand parameters, generally only first-order gradient descent methods are feasible. Higher-order methods are impractical due to their prohibitively high computing complexity. With a compartmentalized learning subunit, calculation of the localized higher-order gradients can be brought into the realm and might help us to train models better and faster. Another potential approach could be building local mechanisms [45] to improve model learning through implicitly utilizing higher-order gradient information.

---

Source code: https://github.com/motifMachine/Dendritic-neural-network

## Acknowledgement

We sincerely thank anonymous reviewers for their insightful comments and suggestions. We also thank Dr. Günter Klambauer for his generous help on the UCI-datasets experiment.

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
