[Supplementary Material]

# Supplementary Material:
# Improved Expressivity Through Dendritic Neural Networks

**Xundong Wu**    **Xiangwen Liu**    **Wei Li**    **Qing Wu**
School of Computer Science and Technology
Hangzhou Dianzi University
Hangzhou, China
wuxundong@gmail.com, wuq@hdu.edu.cn

## A    Schematic drawing of neurons and their dendritic arbors

Figure A1: A schematic drawing of signal transmissions between neurons. It is shown that neurons receive synaptic inputs from proximity neurons on their dendritic arbors. Neurons then process synaptic inputs and send out action potential signals through their axons.

# B  Empirical Measurement of Network Complexity

For a fixed neural network of a certain architecture $A$, inputs $x$ and parameters $W$, our aim is to know how the complexity of the network changes as $A$ changes and across inputs $x$. But it is intractable to go over the entire input space. We study the complexity on a one dimensional *trajectories* through input space as in [2].

**Definition 1** *For two points $x_0, x_1 \in \mathbb{R}^m$, a curve $x(t)$ parameterized by $t \in [0, 1]$, with $x(0) = x_0$ and $x(1) = x_1$ is called a trajectory between $x_0$ and $x_1$.*

If the activation function of a neural network is a piecewise linear function, the function it represents is also a piecewise linear function. And the number of linear pieces (regions) measures the complexity of a model.

**Definition 2** *For a certain $W$, as inputs move from $x$ to $x + \delta$, the piecewise linear activation function of a neuron switches into a different linear region. Then we define this is a neuron transition between $x$ and $x + \delta$.*

A transition of a neuron with ReLU activation function would be given by a neuron switching from on to off (or vice versa) and for Maxout and DENNs by switching index of the maximum input branch. In this way we get the number of transitions undergone by output neurons as we sweep across any given input trajectory $x(t)$.

As in [2], we measure the complexity of the network by the number of neuron transitions. We empirically measure the number of transitions as we sweep $x$ along $x(t)$ to learn its behavior and $x(t) = \cos(\pi t/2)x_0 + \sin(\pi t/2)x_1$, where $x_0, x_1$ are two different data points selected randomly from the MNIST dataset. The ReLU FNNs, DENNs, Maxout networks used in this study have two layers (The output layers for those networks are omitted since they don't use piece-wise activation functions.). For DENNs the branch number are set to $2^1, 2^2, \cdots, 2^8$ denoted by $d$. For the Maxout network, by increasing the number of kernels in a Maxout unit, the number of units is reduced accordingly to keep the total number of synaptic parameters constant. The network weights are randomly initialized with $\sim \mathcal{N}(0, 0.01)$.

As shown in Fig. A2, the transition counts of Maxout networks and DENNs grow as we shift toward larger branch numbers $d$. The transition counts of DENNs with any branch numbers are larger than that of Maxout networks and ReLU FNNs.

Figure A2: The transition counts of Maxout networks and DENNs with different branch numbers and the baseline ReLU FNN.

# C    Additional Results on Permutation Invariant Image Datasets

(a)

(b)

Figure A3: Learning curves on Fashion-MNIST dataset. (a) Training loss curves on DENNs with different $d$ and standard ReLU FNNs (BN-ReLU: batch normalization-ReLU; LN-ReLU: layer normalization-ReLU.). (b) Training loss curves on Maxout networks with different $d$ and standard FNNs.

(a)

(b)

Figure A4: Learning curves on CIFAR-10 dataset. (a) Training loss curves on DENNs of different $d$ and standard FNNS. (b) Training loss curves on Maxout networks of different $d$ and standard FNNs.

(a)

(b)

(c)

Figure A5: Effect of varying the branch number and architecture on model behavior on Fashion-MNIST dataset (BN-ReLU: batch normalization-ReLU; LN-ReLU: layer normalization-ReLU.). (a) Training accuracy results. (b) Training loss results. (c) Test accuracy results.

(a)                                                    (b)

Figure A6: Accuracy curves on Fashion-MNIST dataset for the DENN with $d = 16$, and the Maxout network with $d = 2$, along with ReLU FNNs (BN-ReLU: batch normalization-ReLU; LN-ReLU: layer normalization-ReLU.). (a) Training accuracy curves. (b) Validation accuracy curves

(a)                                                    (b)

(c)

Figure A7: Effect of varying the branch number and architecture on model behavior on CIFAR-10 dataset (BN-ReLU: batch normalization-ReLU; LN-ReLU: layer normalization-ReLU.). (a) Best Training accuracy results. (b) Best training loss results. (c) Test accuracy results.

(a)                                                    (b)

(c)

Figure A8: Effect of varying the branch number and architecture on model behavior on CIFAR-100 dataset (BN-ReLU: batch normalization-ReLU; LN-ReLU: layer normalization-ReLU.). (a) Best training accuracy results. (b) Best training loss results. (c) Test accuracy results.

(a)                                                    (b)

Figure A9: Accuracy curves on CIFAR-10 dataset for the DENN with $d = 16$ and the Maxout network with $d = 2$ along with ReLU FNNs (BN-ReLU: batch normalization-ReLU; LN-ReLU: layer normalization-ReLU.). (a) Training accuracy curves. (b) Validation accuracy curves

(a)                                                    (b)

(c)

Figure A10: Effect of varying the branch number and architecture on model behavior on FASHION-MNIST dataset (BN-ReLU: batch normalization-ReLU; LN-ReLU: layer normalization-ReLU.). DENNs here use a ReLU followed by an average-pooling instead of the Maxout for dendritic nonlinearity. (a) Training accuracy results. (b)Training loss results. (c) Test accuracy results.

(a)

(b)

(c)

Figure A11: Effect of varying the branch number and architecture on model behavior on CIFAR-10 dataset (BN-ReLU: batch normalization-ReLU; LN-ReLU: layer normalization-ReLU.). DENNs here use a ReLU followed by an average-pooling instead of the Maxout for dendritic nonlinearity. (a) Training accuracy results. (b)Training loss results. (c) Test accuracy results.

(a)

(b)

(c)

Figure A12: Effect of varying the branch number and architecture on model behavior on FASHION-MNIST dataset (BN-ReLU: batch normalization-ReLU; LN-ReLU: layer normalization-ReLU.). DENNs here use two ReLU nonlinearties instead of the Maxout for dendritic nonlinearity. (a) Training accuracy results. (b)Training loss results. (c) Test accuracy results.

Figure A13: Effect of varying the branch number and architecture on model behavior on CIFAR-10 dataset (BN-ReLU: batch normalization-ReLU; LN-ReLU: layer normalization-ReLU.). DENNs here use two ReLU nonlinearties instead of the Maxout for dendritic nonlinearity. (a) Training accuracy results. (b)Training loss results. (c) Test accuracy results.

Figure A14: Effect of varying the branch number and architecture on model behavior on random Gaussian noise data. (a) Training accuracy results. (b) Training loss results.

# D Additional Results on 121 UCI Classfication Datasets

Table A1: Test accuracy of DENNs and SNNs, on every prediction task of 121 UCI datasets. The first column lists the name of the datasets, the second column and the third column are the number of samples N and the number of features M, the other columns are the test accuracy of DENNs, SNNs result from [1], and results from our implementation of SNNs. No significant difference is found between the result from our implementation of SNNs and that of the original authors.

| Dataset | N | M | DENN | SNN | SNN-our |
|---|---|---|---|---|---|
| abalone | 4177 | 8 | 0.6638 | 0.6657 | 0.6676 |
| acute-inflammation | 120 | 6 | 1 | 1 | 1 |
| acute-nephritis | 120 | 6 | 1 | 1 | 1 |
| adult | 48842 | 14 | 0.848 | 0.8476 | 0.8508 |
| annealing | 898 | 31 | 0.75 | 0.76 | 0.35 |
| arrhythmia | 452 | 262 | 0.6726 | 0.6549 | 0.6106 |
| audiology-std | 196 | 59 | 0.76 | 0.8 | 0.8 |
| balance-scale | 625 | 4 | 0.9808 | 0.9231 | 0.9679 |
| balloons | 16 | 4 | 1 | 1 | 1 |
| bank | 4521 | 16 | 0.8965 | 0.8903 | 0.8912 |
| blood | 748 | 4 | 0.7326 | 0.7701 | 0.738 |
| breast-cancer | 286 | 9 | 0.6901 | 0.7183 | 0.662 |
| breast-cancer-wisc | 699 | 9 | 0.9771 | 0.9714 | 0.9829 |
| breast-cancer-wisc-diag | 569 | 30 | 0.9859 | 0.9789 | 0.9648 |
| breast-cancer-wisc-prog | 198 | 33 | 0.7143 | 0.6735 | 0.7347 |
| breast-tissue | 106 | 9 | 0.6538 | 0.7308 | 0.6923 |
| car | 1728 | 6 | 0.9884 | 0.9838 | 0.9907 |
| cardiotocography-10clases | 2126 | 21 | 0.823 | 0.8399 | 0.8305 |
| cardiotocography-3clases | 2126 | 21 | 0.9435 | 0.9153 | 0.9322 |
| chess-krvk | 28056 | 6 | 0.8041 | 0.8805 | 0.8794 |
| chess-krvkp | 3196 | 36 | 0.9962 | 0.9912 | 0.9962 |
| congressional-voting | 435 | 16 | 0.578 | 0.6147 | 0.6055 |
| conn-bench-sonar-mines-rocks | 208 | 60 | 0.8269 | 0.7885 | 0.75 |
| conn-bench-vowel-deterding | 990 | 11 | 0.9935 | 0.9957 | 0.9935 |
| connect-4 | 67557 | 42 | 0.8646 | 0.8807 | 0.8799 |
| contrac | 1473 | 9 | 0.5489 | 0.519 | 0.5598 |
| credit-approval | 690 | 15 | 0.8256 | 0.843 | 0.8314 |
| cylinder-bands | 512 | 35 | 0.7812 | 0.7266 | 0.75 |
| dermatology | 366 | 34 | 0.978 | 0.9231 | 0.956 |
| echocardiogram | 131 | 10 | 0.8788 | 0.8182 | 0.8485 |
| ecoli | 336 | 7 | 0.8571 | 0.8929 | 0.8452 |
| energy-y1 | 768 | 8 | 0.9583 | 0.9583 | 0.9583 |
| energy-y2 | 768 | 8 | 0.9062 | 0.9063 | 0.9219 |
| fertility | 100 | 9 | 0.88 | 0.92 | 0.88 |
| flags | 194 | 28 | 0.5208 | 0.4583 | 0.4375 |
| glass | 214 | 9 | 0.6038 | 0.7358 | 0.6038 |
| haberman-survival | 306 | 3 | 0.6579 | 0.7368 | 0.75 |
| hayes-roth | 160 | 3 | 0.8571 | 0.6786 | 0.7857 |
| heart-cleveland | 303 | 13 | 0.5789 | 0.6184 | 0.6053 |
| heart-hungarian | 294 | 12 | 0.7808 | 0.7945 | 0.7808 |
| heart-switzerland | 123 | 12 | 0.4839 | 0.3548 | 0.5161 |
| heart-va | 200 | 12 | 0.32 | 0.36 | 0.32 |
| hepatitis | 155 | 19 | 0.7949 | 0.7692 | 0.7179 |
| hill-valley | 1212 | 100 | 0.5462 | 0.5248 | 0.5066 |
| horse-colic | 368 | 25 | 0.8235 | 0.8088 | 0.8088 |
| ilpd-indian-liver | 583 | 9 | 0.7192 | 0.6986 | 0.7055 |
| image-segmentation | 2310 | 18 | 0.9057 | 0.9114 | 0.8967 |
| ionosphere | 351 | 33 | 0.9659 | 0.8864 | 0.8864 |
| iris | 150 | 4 | 1 | 0.973 | 0.973 |
| led-display | 1000 | 7 | 0.76 | 0.764 | 0.776 |

| | | | | | |
|---|---|---|---|---|---|
| lenses | 24 | 4 | 0.6667 | 0.6667 | 0.6667 |
| letter | 20000 | 16 | 0.962 | 0.9726 | 0.9766 |
| libras | 360 | 90 | 0.7778 | 0.7889 | 0.7778 |
| low-res-spect | 531 | 100 | 0.9023 | 0.8571 | 0.8872 |
| lung-cancer | 32 | 56 | 0.625 | 0.625 | 0.25 |
| lymphography | 148 | 18 | 0.9459 | 0.9189 | 0.9459 |
| magic | 19020 | 10 | 0.8681 | 0.8692 | 0.865 |
| mammographic | 961 | 5 | 0.8083 | 0.825 | 0.825 |
| miniboone | 130064 | 50 | 0.933 | 0.9307 | 0.9293 |
| molec-biol-promoter | 106 | 57 | 0.8846 | 0.8462 | 0.7308 |
| molec-biol-splice | 3190 | 60 | 0.8545 | 0.9009 | 0.857 |
| monks-1 | 556 | 6 | 0.8171 | 0.7523 | 0.8819 |
| monks-2 | 601 | 6 | 0.6505 | 0.5926 | 0.4861 |
| monks-3 | 554 | 6 | 0.8009 | 0.6042 | 0.706 |
| mushroom | 8124 | 21 | 1 | 1 | 1 |
| musk-1 | 476 | 166 | 0.8992 | 0.8739 | 0.8908 |
| musk-2 | 6598 | 166 | 0.9927 | 0.9891 | 0.9915 |
| nursery | 12960 | 8 | 1 | 0.9978 | 0.9981 |
| oocytes_merluccius_nucleus_4d | 1022 | 41 | 0.8392 | 0.8235 | 0.8471 |
| oocytes_merluccius_states_2f | 1022 | 25 | 0.9294 | 0.9529 | 0.8863 |
| oocytes_trisopterus_nucleus_2f | 912 | 25 | 0.8246 | 0.7982 | 0.7851 |
| oocytes_trisopterus_states_5b | 912 | 32 | 0.9474 | 0.9342 | 0.9342 |
| optical | 5620 | 62 | 0.9638 | 0.9711 | 0.9716 |
| ozone | 2536 | 72 | 0.9748 | 0.97 | 0.9716 |
| page-blocks | 5473 | 10 | 0.9613 | 0.9583 | 0.9627 |
| parkinsons | 195 | 22 | 0.8571 | 0.898 | 0.8571 |
| pendigits | 10992 | 16 | 0.9737 | 0.9706 | 0.9714 |
| pima | 768 | 8 | 0.6979 | 0.7552 | 0.7552 |
| pittsburg-bridges-MATERIAL | 106 | 7 | 0.9231 | 0.8846 | 0.8462 |
| pittsburg-bridges-REL-L | 103 | 7 | 0.7308 | 0.6923 | 0.6923 |
| pittsburg-bridges-SPAN | 92 | 7 | 0.7391 | 0.6957 | 0.6957 |
| pittsburg-bridges-T-OR-D | 102 | 7 | 0.84 | 0.84 | 0.84 |
| pittsburg-bridges-TYPE | 105 | 7 | 0.5769 | 0.6538 | 0.6154 |
| planning | 182 | 12 | 0.6 | 0.6889 | 0.6889 |
| plant-margin | 1600 | 64 | 0.8325 | 0.8125 | 0.81 |
| plant-shape | 1600 | 64 | 0.725 | 0.7275 | 0.75 |
| plant-texture | 1599 | 64 | 0.81 | 0.8125 | 0.815 |
| post-operative | 90 | 8 | 0.6818 | 0.7273 | 0.6818 |
| primary-tumor | 330 | 17 | 0.5366 | 0.5244 | 0.5244 |
| ringnorm | 7400 | 20 | 0.9757 | 0.9751 | 0.9795 |
| seeds | 210 | 7 | 0.9231 | 0.8846 | 0.9423 |
| semeion | 1593 | 256 | 0.9673 | 0.9196 | 0.9296 |
| soybean | 683 | 35 | 0.8803 | 0.8511 | 0.8936 |
| spambase | 4601 | 57 | 0.9487 | 0.9409 | 0.9348 |
| spect | 265 | 22 | 0.6237 | 0.6398 | 0.6398 |
| spectf | 267 | 44 | 0.893 | 0.4973 | 0.5401 |
| statlog-australian-credit | 690 | 14 | 0.6105 | 0.5988 | 0.6395 |
| statlog-german-credit | 1000 | 24 | 0.72 | 0.756 | 0.744 |
| statlog-heart | 270 | 13 | 0.9254 | 0.9254 | 0.8955 |
| statlog-image | 2310 | 18 | 0.9775 | 0.9549 | 0.9532 |
| statlog-landsat | 6435 | 36 | 0.899 | 0.91 | 0.912 |
| statlog-shuttle | 58000 | 9 | 0.9991 | 0.999 | 0.9989 |
| statlog-vehicle | 846 | 18 | 0.8104 | 0.8009 | 0.8009 |
| steel-plates | 1941 | 27 | 0.7753 | 0.7835 | 0.7732 |
| synthetic-control | 600 | 60 | 0.9933 | 0.9867 | 0.9933 |
| teaching | 151 | 5 | 0.5789 | 0.5 | 0.6053 |
| thyroid | 7200 | 21 | 0.9822 | 0.9816 | 0.977 |
| tic-tac-toe | 958 | 9 | 0.9833 | 0.9665 | 0.9833 |
| titanic | 2201 | 3 | 0.7873 | 0.7836 | 0.7836 |

| | | | | | |
|---|---|---|---|---|---|
| trains | 10 | 29 | NA | NA | NA |
| twonorm | 7400 | 20 | 0.9816 | 0.9805 | 0.9795 |
| vertebral-column-2clases | 310 | 6 | 0.8571 | 0.8312 | 0.8442 |
| vertebral-column-3clases | 310 | 6 | 0.8052 | 0.8312 | 0.8442 |
| wall-following | 5456 | 24 | 0.9186 | 0.9098 | 0.9091 |
| waveform | 5000 | 21 | 0.8392 | 0.848 | 0.8792 |
| waveform-noise | 5000 | 40 | 0.8432 | 0.8608 | 0.852 |
| wine | 178 | 13 | 1 | 0.9773 | 0.9545 |
| wine-quality-red | 1599 | 11 | 0.635 | 0.63 | 0.645 |
| wine-quality-white | 4898 | 11 | 0.6225 | 0.6373 | 0.6242 |
| yeast | 1484 | 8 | 0.5822 | 0.6307 | 0.6065 |
| zoo | 101 | 16 | 1 | 0.92 | 0.96 |