[Reviews · NeurIPS 2018]

Reviewer 1



[Remarks following on the authors' response are included below in brackets.] This paper presents D-Nets, an architecture loosely inspired by the dendrites of biological neurons. In a D-Net, each neuron receives input from the previous layer as the maxpool of linear combinations of disjoint random subsets of that layer’s outputs. The authors show that this approach outperforms self-normalizing neural networks and other advanced approaches on the UCI collection of datasets (as well as outperforming simple non-convolutional approaches to MNIST and CIFAR). They provide an intuition that greater fan-in to non-linearities leads to a greater number of linear regions and thus, perhaps, greater expressibility. I am still quite surprised that such a simple method performs so well, but the experimental setup seems sound. I would ideally like a further analysis of the behavior of D-Nets. For example, how does the optimal number of branches grow with the size of the layer? [The authors note in their response that no pattern was observed in preliminary experiments on this question. This negative result would be interesting to note in the paper.] It would also be interesting to see an analysis of other nonlinear functions than maxpool in combining the contributions from different dendritic branches. [The authors note in their response that maxpool was the most effective nonlinearity tested. Other nonlinearities tested should perhaps be noted in the paper, even if they gave weaker results.] The main theoretical result, Theorem 3, shows that D-Nets are universal approximators. This follows from a simple proof using the facts that (i) any piecewise linear (PWL) function is the difference of two convex PWL functions and (ii) any convex PWL function can be expressed as the max of line segments, and is thus expressible as a D-Net. D-Nets are a type of feedforward neural net (FNN) (with sparsity and max-pooling), but in various places, the paper compares FNNs to D-Nets as if they were two separate things. For example, the definition reads: “In comparison, a dendritic neural network…” This terminology should be changed - for example, to MLP in place of FNN, or simply by saying “standard FNN”. p. 3: The definition of k seems unnecessarily complicated and also slightly contradictory, since it can vary by 1 between different branches. One might simply write k = n_{l – 1} / d, eliding the discussion of floors (which probably don’t matter in general unless k is very small). Figure 2 could be more extensively labeled, showing d_1 = 1, n_1, etc. Lemma 1 should be presented in such a way that it is clearer it was proven in [32]. E.g. “Lemma 1 (Wang [32]).” Regarding compositionality and expressibility, references could be added also to Mhaskar, Liao, Poggio (2017) and Rolnick, Tegmark (2018). Line 171: The function T_A is never actually defined rigorously, and this notation should perhaps simply be omitted since it is not used again. The “expressivity” of a network is not a quantity – or rather, it can be measured by many quantities. It seems that here the authors are interested in the number of linear regions of a piecewise linear function. It should be made clearer that no statements about T_A are actually proven – merely tested empirically. There are numerous typos and small grammar issues throughout the paper, which detract from its readability. These should be corrected by careful proofreading. Some examples from just the first page: - ln 5: “accumulator” should be “accumulators”. - ln 9: “non-linearity” should be “non-linearities”. - ln 12: “significant boost” should be “a significant boost”. - ln 12: “its generalization ability” should be “their generalization ability”. - ln 13: “in comparison” should be omitted. - ln 18: “made into” should be omitted. - ln 20: “as simple weighted sum” should be “as a simple weighted sum”.

Reviewer 2



[I include in square brackets, below in my point-by-point criticisms, my responses to the author's feedback on the review. Overall, my opinion has not changed: this is a strong submission.] The authors build a new neural network architecture (D-nets) in which inputs impinge on different dendritic branches, each of which takes a weighted some over those inputs. The unit output is the max over the activations of its dendritic branches. This is a variant on the maxout networks, which have a similar architecture. The key difference appears to be that the D-net dendritic branches get sparse non-overlapping sets of inputs, whereas in maxout, each branch gets all of the inputs. The authors use a similar proof strategy to Ref. 6, to show that D-nets can be universal function approximators, and evaluate both the expressivity of their networks (using transition number as the measure), and the performance of their networks on a wide range of ML tasks. Performance is compared with maxout, and with traditional neural nets, and the D-nets are found to have the highest expressivity and best performance of the architectures considered. Overall, I was impressed by this submission. The comparisons to previous network structures are very thorough, the performance appears to be quite good, the D-net idea is straightforward but (as far as I can tell) somewhat novel, and the paper is relatively well-written. I have a few comments / suggestions, listed below. 1. The proof in Section 3.2 is nice, but a little restrictive. It would be nice to have a more general proof, applying to conditions other than d_1=1. If such a proof is possible, I would highly recommend including it. Also, if there are conditions on d such that the networks are not universal function approximators, those conditions should be identified and stated. [I am glad to hear that the proof can be generalized beyond d_1=1. I am somewhat concerned about the need for non-random connectivity that the first layer: is the requirement very restrictive? It would be worth clarifying that requirement (when it is likely to be satisfied or not), so that readers know clearly when they should, or should not, expect the DNNs to be universal function approximators.] 2. In the discussion of Sparse neural networks, there are a few recent neuroscience papers that are relevant, and that the authors should consider reading and/or citing (listed below). These might inform the optimal "d" values of the D-networks, and the methods from these papers could be relevant for understanding the optimal "d" values in the D-nets. a) Litwin-Kumar, A., Harris, K.D., Axel, R., Sompolinsky, H. and Abbott, L.F., 2017. Optimal degrees of synaptic connectivity. Neuron, 93(5), pp.1153-1164. b) Cayco-Gajic, N.A., Clopath, C. and Silver, R.A., 2017. Sparse synaptic connectivity is required for decorrelation and pattern separation in feedforward networks. Nature Communications, 8(1), p.1116. [I am glad that you found these useful.] 3. It took me awhile to figure out that the acronyms "BN" and "LN" in the figures and captions were batch norm and layer norm. I would suggest spelling those out in the legend to Fig. 4. [Thanks for adding these; I think they will help with clarity.] 4. Why give the Avg. rank diff. instead of just Avg. rank? As far as I can tell, these diff values are just avg. rank - 4.5, and reporting them makes the actual rank values less transparent. [It sounds like you will change these to avg. rank, which I support. I understand that there is precedent for the other way of reporting the results (avg. rank diff.), but sometimes previous work was flawed, and it's best not to repeat those flaws...]

Reviewer 3



UPDATED REVIEW: the authors responded to most comments. Although they provided details in the response, It's not clear to me if they will include discussions of (1) piecewise activation functions and in neuroscience e.g. two-compartment models in the related work (2) generalization (+learning curves) in the manuscript; I think both these things would improve the work. I still find it strange that no generalization results were presented in the original submission. I agree with the other reviewers that the title sounds strange (typically you're advised not to use subjective descriptions like "great" or "amazing" in scientific work). I have updated my rating to 6. ======================== PAPER SUMMARY:The authors propose a network architecture to model synaptic plasticity in dendritic arbours, which ends up looking to me like alternating layers of maxout and regular units, with some randomly-dropped connections. The proposed architecture is tested on MNIST, CIFAR, and a range of UCI classification tasks, and found to outperform alternatives. Insight about why this is the case is provided in the form of an analysis of expressivity, measured as number of linear regions in loss space. STRENGTHS: - quality: The paper is generally well-written and the structure and experiments are largely well-suited to evaluate the proposed architecture. The introduction and related work do a good job contextualizing the work. The experiments about expressivity are interesting and relevant. - clarity: What is proposed and what is done are both clear. Details of datasets, experiments, and explanations of related work are generally very good. - originality: To my knowledge, the proposed architecture is novel. Being somewhat familiar with the biology, I find piecewise linear/max operations a reasonable approximation intuitively. WEAKNESSES: - quality: It seems to me that the chosen "algorithm" for choosing dendrite synapses is very much like dropout with a fixed mask. Introducing this sparsity is a form of regularization, and a more fair comparison would be to do a similar regularization for the feed-forward nets (e.g. dropout, instead of bn/ln; for small networks like this as far as I know bn/ln are more helpful for optimization than regularization). It also seems to me that the proposed structure is very much like alternating layers of maxout and regular units, with this random-fixed dropout; I think this would be worth comparing to. I think there are some references missing, in the area of similar/relevant neuroscience models and in the area of learned piecewise activation functions. It would be reassuring to mention the computation time required and whether this differs from standard ff nets. Also, most notably, there are no accuracy results presented, no val/test results, and no mention is made of generalization performance for the MNIST/CIFAR experiments. - clarity: Some of the sentences are oddly constructed, long, or contain minor spelling and grammar errors .The manuscript should be further proofread for these issues. For readers not familiar with the biological language, it would be helpful to have a diagram of a neuron/dendritic arbour; in the appendix if necessary. It was not 100% clear to me from the explanations whether or not the networks compared have the same numbers of parameters; this seems like an important point to confirm. - significance: I find it hard to assess the significance without generalizaion/accuracy results for the MNIST/CIFAR experiments. REPRODUCABILITY: For the most part the experiments and hyperparameters are well-explained (some specific comments below), and I would hope the authors would make their code available. SPECIFIC COMMENTS: - in the abstract, I think it should say something like "...attain greater expressivity, as measured by the change in linear regions in output space after [citation]. " instead of just "attain greater expressivity" - it would be nice to see learning curves for all experiments, at least in an appendix. - in Figure 1, it would be very helpful to show a FNN and D-Net with the same number of parameters in each (unless I misunderstood, the FNN has 20 and the DNN has 16). - There are some "For the rest part" -> for the rest of (or rephrase) - missing references: instead of having a section just about Maxout networks, I think the related work should have a section called something like "learned piecewise-linear activation functions" which includes maxout and other works in this category, e.g. Noisy Activation Functions (Gulcehre 2016). Also, it's not really my field but I believe there is some work on two-compartment models in neuroscience and modeling these as deep nets which would be quite relevant for this work. - It always bothers me somewhat when people refer to the brain as 'sparse' and use this as a justification for sparse neural networks. Yes, overall/considering all neurons the brain as one network it would be sparse, but building a 1000 unit network to do classification is much more analogous to a functional subunit of the brain (e.g. a subsection of the visual pathway), and connections in these networks are frequently quite dense. The authors are not the first to make this argument and I am certainly not blaming them for its origin, but I take this opportunity to point it out as (I believe) flawed. :) - the definition of "neuron transition" is not clear to me - the sentence before Definition 2 suggests that it is a change in _classification_ (output space), which leads to a switch in the linear region of a piecewise linear function, but the Definition and the subsequent sentence seem to imply it is only the latter part (a change from one linear region of the activation function to another; nothing to do with the output space). If the latter, it is not clear to me how/whether or not these "transitions" say anything useful about learning. If it is the former (more similar to Raghu et al), I find the definition given unclear. - I like the expressiveness experiments, but It would be nice to see some actual numbers instead of just descriptions. - unless I missed it somehow, the "SNN" is never defined. and it is very unclear to me whether it refers to a self-organizing neural network cited in [12]. or a "sparse" neural network, and in any case what exactly this architecture is. - also possible I missed it despite looking, but I could not find what non-linearity is used on D-Nets for the non-dendrite units OVERALL ASSESSMENT: My biggest issue with the paper is the lack of mention/results about generalization on MNIST/CIFAR, and the ambiguity about fair comparison. If these issues are resolved I would be very willing to change my rating. CONFIDENCE IN MY SCORE: This is the first time I've given a confidence of 5. With due credit to the authors, I believe I've understood most things about the paper, and I am familiar with the relevant work. Of course I'm not infallible and it's possible I've missed or misunderstood something, especially relating to the things I noted finding unclear.